# Out-of-pocket expenditures and financial risks associated with treatment of vaccine-preventable diseases in Ethiopia: A cross-sectional costing analysis

**Solomon Tessema Memirie** [1] *, **Mieraf Taddesse Tolla**[1], **Eva Rumpler**[2], **Ryoko Sato**[3], **Sarah Bolongaita**[3], **Yohannes Lakew Tefera**[4], **Latera Tesfaye**[5], **Meseret Zelalem Tadesse**[4], **Fentabil Getnet**[5], **Tewodaj Mengistu**[6], **Stéphane Verguet**[3]

**1** Addis Center for Ethics and Priority Setting, College of Health Sciences, Addis Ababa University, Addis Ababa, Ethiopia, **2** Department of Epidemiology, Harvard T.H. Chan School of Public Health, Boston, Massachusetts, United States of America, **3** Department of Global Health and Population, Harvard T.H. Chan School of Public Health, Boston, Massachusetts, United States of America, **4** Ministry of Heath of Ethiopia, Addis Ababa, Ethiopia, **5** National Data Management Center for Health, Ethiopian Public Health Institute, Addis Ababa, Ethiopia, **6** Gavi, the Vaccine Alliance, Geneva, Switzerland

* tess_soul@yahoo.com

**Data Availability Statement:** The link to the public data repository is available at: https://doi.org/10.6084/m9.figshare.20425434.

**Funding:** This work was supported by Gavi, the Vaccine Alliance (grant to SV). The funder had no

## Abstract

### Background

Vaccine-preventable diseases (VPDs) remain major causes of morbidity and mortality in low- and middle-income countries (LMICs). Universal access to vaccination, besides improved health outcomes, would substantially reduce VPD-related out-of-pocket (OOP) expenditures and associated financial risks. This paper aims to estimate the extent of OOP expenditures and the magnitude of the associated catastrophic health expenditures (CHEs) for selected VPDs in Ethiopia.

### Methods and findings

We conducted a cross-sectional costing analysis, from the household (patient) perspective, of care-seeking for VPDs in children aged under 5 years for pneumonia, diarrhea, measles, and pertussis, and in children aged under 15 years for meningitis. Data on OOP direct medical and nonmedical expenditures (2021 USD) and household consumption expenditures were collected from 995 households (1 child per household) in 54 health facilities nationwide between May 1 and July 31, 2021. We used descriptive statistics to measure the main outcomes: magnitude of OOP expenditures, along with the associated CHE within households. Drivers of CHE were assessed using a logistic regression model. The mean OOP expenditures per disease episode for outpatient care for diarrhea, pneumonia, pertussis, and measles were $5·6 (95% confidence interval (CI): $4·3, 6·8), $7·8 ($5·3, 10·3), $9·0 ($6·4, 11·6), and $7·4 ($3·0, 11·9), respectively. The mean OOP expenditures were higher for inpatient care, ranging from $40·6 (95% CI: $12·9, 68·3) for severe measles to $101·7 ($88·5, 114·8) for meningitis. Direct medical expenditures, particularly drug and supply expenses, were the major cost drivers.

role in study design, data collection and analysis, decision to publish, or preparation of the manuscript.

**Competing interests:** The authors have declared that no competing interests exist.

**Abbreviations:** AE, adult equivalent; BCG, bacille Calmette-Guérin; CBHI, community-based health insurance; CHE, catastrophic health expenditure; CI, confidence interval; DME, direct medical expenditure; DNME, direct nonmedical expenditure; DOVE, Decade of Vaccine Economics; DPT, Diphtheria-Pertussis-Tetanus; GNI, gross national income; GVAP, Global Vaccine Action Plan; Hib, *Haemophilus influenzae* type B; IHE, impoverishing health expenditure; IMCI, integrated management of childhood illnesses; LMIC, low- and middle-income country; OOP, out-of-pocket; PHC, primary healthcare; PPP, purchasing power parity; SD, standard deviation; SDG, Sustainable Development Goal; SNNP, Southern Nations, Nationalities, and Peoples; UHC, universal health coverage; VPD, vaccine-preventable disease.

Among those who sought inpatient care (345 households), about 13·3% suffered CHE, at a 10% threshold of annual consumption expenditures. The type of facility visited, receiving inpatient care, and wealth were significant predictors of CHE (*p*-value < 0·001) while adjusting for area of residence (urban/rural), diagnosis, age of respondent, and household family size. Limitations include inadequate number of measles and pertussis cases.

## Conclusions

The OOP expenditures induced by VPDs are substantial in Ethiopia and disproportionately impact those with low income and those requiring inpatient care. Expanding equitable access to vaccines cannot be overemphasized, for both health and economic reasons. Such realization requires the government's commitment toward increasing and sustaining vaccine financing in Ethiopia.

## Author summary

### Why was this study done?

- Despite a rapid expansion in access to vaccines in the past 2 decades, vaccine-preventable diseases (VPDs) remain major causes of morbidity and mortality in low- and middle-income countries.

- Out-of-pocket (OOP) medical expenditures can lead to catastrophic health expenditures and impoverishment.

- Studies on household healthcare expenditures and associated financial risks for VPDs among children in sub-Saharan African countries are scarce.

### What did the researchers do and find?

- We collected OOP expenditures data from 995 households to estimate medical impoverishment associated with the following vaccine-preventable childhood diseases: measles, pertussis, pneumonia, diarrhea, and meningitis.

- Households incur substantial OOP expenditures for the treatment of VPDs in Ethiopia.

- Poor families and those with sick children requiring inpatient care are likely to be impoverished.

### What do these findings mean?

- Expanding access to vaccination has the potential to protect families from OOP expenditures related to the treatment of VPDs and its associated catastrophic and impoverishing financial consequences. The financial risk benefits primarily accrue among the poorest.

- Universal access to vaccines requires the government's commitment toward increasing and sustaining vaccine financing.

## Introduction

Vaccine-preventable diseases (VPDs) such as pneumonia, whooping cough (pertussis), diarrhea, measles, and meningitis are among the major causes of child morbidity and mortality in low- and middle-income countries (LMICs) [1]. In the past 2 decades, access to vaccines has seen rapid expansion in LMICs with substantial reductions in mortality [2]. It is estimated that vaccination against measles, *Haemophilus influenzae* type B (Hib), *Streptococcus pneumoniae*, rotavirus, and *Neisseria meningitidis* serogroup A would avert nearly 20 million deaths between the years 2020 and 2030 in LMICs [2].

Immunization coverage has progressively increased in Ethiopia, a low-income country with Africa's second largest population, but still large coverage gaps have remained [3,4]. For instance, in 2019, coverage of pentavalent 3 (Diphtheria-Pertussis-Tetanus (DPT), Hib and hepatitis B virus, third dose), first dose of measles vaccine (MCV), and full immunization (bacille Calmette-Guérin (BCG), polio, DPT, and MCV) were only 61%, 59%, and 43%, respectively [5]. WHO and UNICEF together launched the Global Vaccine Action Plan (GVAP) 2011–2020 that urged for a national coverage of 90% of DPT third dose (DPT3) in at least 80% of districts in LMICs [6]. Furthermore, the new immunization commitments as part of the Immunization Agenda 2030 have a strong focus on equity through extending immunization services to regularly reach "zero-dose" and underimmunized children at country, regional, and global levels by 2030 [7]. The gaps in immunization coverage and the corresponding VPD burden disproportionately affect the rural populations, the less educated, and those most economically deprived [4,5]. Therefore, Ethiopia fell short of its commitment to GVAP targets that would have otherwise contributed to large reductions in VPD burden in the country. Ethiopia was among the 5 countries (next to the Democratic Republic of the Congo, Nigeria, India, and Pakistan) with the largest number of unprotected infants as of 2018 [7]. As a result, VPDs remain one of the commonest childhood illnesses in the country with, for instance, recurrent measles outbreaks, and with pneumonia, diarrhea, measles, pertussis, and meningitis estimated to account for approximately 11% of total deaths in 2019 [8,9]. Furthermore, these VPDs often result in large economic losses through increased use of sparse healthcare resources and productivity decreases [10,11].

While vaccines are among the most cost-beneficial public health interventions, the evidence base on the costs of illness averted due to vaccination in low-income countries such as Ethiopia remains limited [12]. A 2013 study focusing on out-of-pocket (OOP) expenditures induced by pneumonia and diarrhea treatment in Ethiopia concluded that households incurred considerable expenditures with significant financial hardship [13]. The Decade of Vaccine Economics (DOVE) project assessed large treatment expenditures and productivity losses due to pneumonia, diarrhea, and measles in 3 countries including the low-income country of Uganda [14]. The OOP expenditure estimates for pneumonia and diarrhea in Ethiopia and Uganda varied by type of facility visited, service type (inpatient versus outpatient care), residence, and household income [13,14].

Illness can impose a substantial financial burden on individuals and households. OOP health-related expenditures (at the point of care) can impede accessing care and impoverish families with "catastrophic" health expenditures (CHEs), that is OOP expenditures surpassing a certain threshold of income or consumption expenditures [15,16]. Aggregate OOP health expenditures contributed to 31% of total health expenditures in Ethiopia in the years 2019/2020, which is greater than the global average (21%) or the 20% level suggested by WHO [17,18]. Furthermore, a study covering more than 30 African countries including Ethiopia reported that underfinancing of immunization programs, vaccine stock-outs, and logistical

supply chain challenges usually drive the prevailing underperformance of immunization programs in these countries [19]. The study reported that the Ethiopian government would only finance about 14% of its immunization program, due to the extreme scarcity of domestic financial resources in the country (about USD 6 government health expenditure per capita in 2019) [3].

Universal access to vaccination would substantially reduce VPD-related OOP expenditures and associated impoverishment, in addition to its large impact on mortality and morbidity, especially so for the most marginalized populations [20]. In short, vaccines could play a major role in the progressive realization of universal health coverage (UHC) and the Sustainable Development Goals (SDGs), specifically regarding SDG-1 ("To end poverty in all its forms everywhere") and SDG-3 ("To ensure healthy lives and promote well-being for all at all ages") targets in Ethiopia [21].

Empowering countries to take ownership of their own immunization programs and achieve financial sustainability are among the long-term goals of Gavi, the Vaccine Alliance. Based on their gross national income (GNI) per capita, countries are expected to allocate an increasing amount of financial resources toward vaccines, eventually facing graduation from Gavi support [22]. Therefore, understanding the full economic burden of VPDs, including documenting the extent of illness-related OOP expenditures, can help set highly effective health policies from both a health and economic standpoint. However, the evidence base to substantiate such arguments remains scarce in Ethiopia.

The objective of this study is to report on the incurred household OOP expenditures and productivity losses due to pneumonia, diarrhea, measles, meningitis, and pertussis, and to assess the extent to which such financial burdens contribute to household medical impoverishment, by socioeconomic group and area of residence in Ethiopia. Better estimates of household OOP expenditures for the treatment of VPDs in LMICs allow for more precision in estimating the expected financial risk protection and return on investment of expanding access to vaccination.

## Methods

We conducted a cross-sectional costing analysis of care-seeking for VPDs, from the household (patient) perspective, in children aged under 5 years for pneumonia, diarrhea, measles, and pertussis, and for children aged under 15 years for meningitis. OOP expenditure data were collected directly in local currency (Ethiopian birr or ETB), and then converted to US dollars (USD). We used the median exchange rate for the data collection period (May 1 to July 31, 2021), that is ETB 43·4 = USD 1 [23]. We also used an exchange rate of ETB 12·1 per unit of purchasing power parity (PPP) $ (year 2020) [24]. The study is reported as per the Strengthening the Reporting of Observational Studies in Epidemiology (STROBE) guideline (S1 STROBE checklist). Furthermore, the data collection and analysis were based on a prospective protocol (S1 PROSPECTIVE protocol).

### Study area and population

Ethiopia is Africa's second most populous country with an estimated population of 121 million in 2022; nearly 47% of its population is under the age of 15 years, and 80% are rural inhabitants [25,26]. It is a low-income country with a 2020 GNI per capita of around $936 and average male and female life expectancies at birth of 65 and 69 years, respectively [3]. At present, Ethiopia is administratively structured into 11 regions (Oromia; Amhara; Southern Nations, Nationalities, and Peoples (SNNP); Sidama; South West Ethiopia Peoples Region; Tigray; Benishangul-Gumuz; Gambella; Afar; Somali; and Harari) and 2 city administrations (Addis

Ababa and Dire Dawa). A network of facilities organized in a 3-tier health system model provides healthcare services in Ethiopia [26]. Its primary healthcare (PHC) unit (with 17,550 health posts and 3,735 health centers) constitutes the first level providing primary care services, especially for rural communities. There are a total of 353 primary, general (secondary level), and specialized (tertiary level) hospitals [26].

## Study sites and participant recruitment

The study participants were individuals seeking treatment services from a sample of 54 systematically selected public health facilities constituted of 18 hospitals, 26 health centers, and 10 health posts. Ethiopian regions displayed wide variations in immunization coverage levels [4]. Therefore, we first selected regions that performed below the national average on DPT3 vaccine, pneumococcal conjugate vaccine (PCV-13), MCV, or full immunization coverage. Hence, Oromia, SNNP, Amhara, Afar, and Somali were selected [27]. Since Addis Ababa serves as a referral destination for the whole Ethiopia, we included 2 randomly selected hospitals there. Based on 2018 population projections, these regions accounted for 91% of the total Ethiopian population and for 93% of under-15-year-olds nationwide. Accordingly, Oromia, Amhara, SNNP, Somali, and Afar had 44%, 22%, 24%, 7%, and 2% of under-15-year-olds, respectively, and thus contributed proportionally to the selection of each facility type [28].

Second, to obtain a sample that would more likely cover high-burden areas and therefore maximize recruitment of adequate numbers of VPD cases within regions, we used 2 datasets. The first dataset was DPT3 coverage disaggregated by zone: These estimates, drawn from the Global Burden of Disease study, showed the percentage of children in each zone that did not receive DPT3, which allowed identifying possible "hot spots" within regions [29]. The second dataset was Ethiopia's master health facility list extracted from the Service Provision Assessment survey conducted in 2014: This provided the list of health facilities (by type, in each region) with respective zonal codes that we could link to the zone-level DPT3 coverage estimates [30]. Hence, we could derive 5 sampling frames of hospitals located within close proximity of the lowest DPT3 coverage zone for each region. Subsequently, we randomly selected the required number of hospitals in each region. We then prepared 5 sampling frames of health centers (per region) by listing all the health centers located in the same town as the hospitals included, and we followed the same random selection process to select the final list of health centers to be included. For the health posts, we selected those posts closest to the already selected health centers to maximize operational efficiency. We included public facilities only because around 75% of care seeking takes place in public facilities (predominantly health centers) in Ethiopia [27].

We included children 0 to 59 months of age who visited the selected health facilities with any of the following conditions: pneumonia, diarrhea, pertussis, or measles. For meningitis, children less than 15 years were included. Based on a previous Ethiopian study for pneumonia and diarrhea (adding here measles and pertussis), we estimated the difference in mean OOP expenditures across any 2 successive wealth quintiles to be USD5 with a standard deviation (SD) of USD10 [13]. For bacterial meningitis, given the possibly long hospital stays associated (a 3-fold increase in mean length of stay for bacterial meningitis as compared to other conditions), the mean difference was assumed to be USD15 (with SD of USD32) [31]. Accordingly, with 80% power to detect such differences with 95% confidence, we would need 66 patients per quintile for meningitis and 63 per quintile for each of the other 4 conditions. Assuming a 5% nonresponse rate, this would mean that we would need to recruit 347 cases of meningitis (all inpatient cases) and 331 cases for pneumonia, diarrhea, measles, and pertussis, respectively (with 10% to 20% of cases from inpatient wards). This would give a total sample size of 1,670 patients.

In each facility, healthcare providers (hired and trained as data collectors for this study) were tasked with proactively identifying eligible patients from pediatric outpatient departments and inpatient wards based on patient diagnosis documented on medical charts. Outpatient cases were selected when a clinician trained on integrated management of childhood illnesses (IMCI) identified them as diarrhea, pneumonia, pertussis, or measles cases until the target sample size was reached. Similarly, severe cases of pneumonia, diarrhea, measles or meningitis were consecutively enrolled from pediatric inpatient units after the clinician in charge had confirmed the diagnosis.

Children less than 5 years of age presenting with either diarrhea, pneumonia, pertussis, or measles and without other concomitant illness were included in the study. Similarly, we included children less than 15 years who had meningitis and without other conditions. Children with multiple conditions and beyond the specified ages were excluded from the study.

## Data collection

We employed face-to-face interviews with the parents/caregivers of ill children at the health facility followed by phone call interviews within 2 to 4 weeks of the initial in-facility interview. Data on direct medical expenditures (DMEs) (e.g., consultation, diagnostic workup, medications, hospital stays), direct nonmedical expenditures (DNMEs) (e.g., transportation, food and drinks, lodging), and caregivers' time losses were collected using a structured questionnaire (S1 Data Tool) adapted from previous studies [13]. Furthermore, during the face-to-face interviews, caregivers were asked whether they had used over-the-counter medications and/or had any visit to any other healthcare facility for the ongoing disease episode. We also collected data on caregivers' (for a maximum of 3) time losses and how those times would have been spent by the caregiver if she/he was not taking care of the sick child. We derived parents' time losses by adding the time spent seeking care prior to outpatient consultation and/or inpatient admission to the duration of outpatient and/or inpatient visit. Additionally, we collected data on the employment status of each caregiver and if there were any losses in income or wage while taking care of the sick child.

Household consumption expenditures data were collected by asking caregivers (when possible household heads) for estimates of their expenses on food, fuel, electricity, water, rent, education, telephone, leisure, and healthcare for the month preceding the interview. The monthly data were then used to estimate annual household expenditures. We also collected information on the household sources of financing used to cope with OOP expenditures.

To facilitate data collection and ensure data quality, 4 research coordinators were hired and trained: one for each of the 3 large regions (Oromia, Amhara, and SNNP), and one for Afar, Somali, and Addis Ababa together. The central coordinator, together with regional coordinators, conducted a pilot testing of the Oromiffa, Amharic, and Somali language versions of the questionnaires. Each regional coordinator travelled to health facilities, identified a nurse, health officer, or medical doctor (for hospitals and health centers) or an IMCI-trained health extension worker (for health posts), and subsequently provided training on the use of data collection tools. The regional coordinators then observed the data collection process and gave feedback on at least 2 patients; they had additional visits to each facility 2 more times to provide onsite support, data quality checks, and eventually collect the completed questionnaires. All data were collected during May 1 to July 31 2021. All study participants gave a written informed consent.

## Data analysis

For each disease episode, we computed the total OOP expenditures as the sum of direct medical and direct nonmedical expenses. For direct medical expenses, we added up OOP payments

for consultation, diagnostic workup, medications, and hospital stay. For direct nonmedical expenses, we added OOP expenditures for transportation, food, lodging, and other expenditures incurred in relation to seeking care. Separately, we computed an economic value for the productivity losses associated with caregivers' time during care seeking. We computed productivity losses for income- or wage-earning family members as the sum of reported income losses of household members who engaged in the care of the sick child, i.e., reported income losses by caregiver-1 plus caregiver-2 plus caregiver-3.

We constructed 5 quintiles using the total household consumption expenditures and an adult equivalent (AE) score (AE is calculated using the formula: $AE = (A + \alpha K)^{\theta}$, where A is the number of adults in the household, K is the number of children, $\alpha$ is the "cost of children," and $\theta$ reflects the degree of economies of scale. On the basis proposed by Deaton and Zaidi, we chose a value of 0.3 for $\alpha$ and 0.9 for $\theta$, because food accounts for a large proportion of total consumption and economies of scale are relatively limited) [32].

We measured the incidence of CHE (i.e., catastrophic payment headcount) associated with OOP expenditures for an episode of pneumonia, diarrhea, measles, pertussis, and meningitis. A case of CHE was calculated by computing OOP expenditures incurred minus any reimbursement from third-party payers divided by annual household expenditures [33,34]. Specifically, a household would incur CHE when OOP expenditures exceeded a certain threshold (10% as commonly used) of consumption expenditures [15]. Additionally, we computed CHE cases using household nonfood expenditures (capacity to pay defined as effective income net of subsistence spending) as the denominator and a 40% catastrophic threshold [15,35]. The thresholds represent severe disruptions in customary living standards of families as a result of healthcare spending. As the diseases considered were acute conditions, we examined both short- and long-term impact, using either monthly or annual household consumption expenditures as the denominator in the computation of CHE cases. We also conducted a sensitivity analysis using a lower (5%) threshold of annual consumption expenditures.

Furthermore, we estimated the proportion of households facing impoverishing health expenditures (IHE) from OOP expenditures (using monthly consumption expenditures). An IHE case was counted when a household would fall below the poverty line with incurred OOP expenditures. The poverty headcount was the fraction of people living in poverty. We used the World Bank definition for extreme poverty (for low-income countries) as those living on less than $1·90 a day (in 2020 PPP), which was converted into a poverty line (per AE per year) of ETB 8401 [24].

Lastly, we determined whether household socioeconomic status, type of health facility visited, hospitalization, region, and residence (urban versus rural) were associated with the extent of OOP expenditures incurred. To adjust for the skewness of the distribution in OOP expenditures, we used a logarithmic transformation. We then ran linear regression models to assess the level of variation in the magnitude of OOP expenditures by wealth quintile, area of residence, facility type, and region: Logistic regressions were conducted to identify the drivers of differences in the odds of CHE, with firstly implementing bivariate models, and, secondly, multivariate models.

All analyses were conducted with STATA (version 17).

## Ethical clearance

The study was reviewed and approved by the Scientific Ethical Review Committee of the Ethiopian Public Health Institute (EPHI-IRB-255-2020), the Institutional Review Board (IRB) of Saint Paul's Hospital Millennium Medical College (pm23/682), and the IRB of the Harvard T. H. Chan School of Public Health (IRB20-0285).

## Results

### Sociodemographic characteristics

Data on OOP expenditures were collected from 54 public health facilities (10 health posts, 26 health centers, and 18 hospitals) from Oromia, SNNP, Amhara, Somali, and Afar regions and Addis Ababa city administration. A total of 995 patients (1 child per household) with a mean age of 1.9 years (95% confidence interval (CI): 1.7 to 2.0 years) were included in the final analysis, of whom 411 (41%) were rural residents (Table 1). We conducted follow-up phone calls after a mean duration of 14.4 days (SD: 0.8 days and range of 11 to 21 days) following the face-to-face interviews. All the households except one (994 out of 995) responded to the follow-up phone calls. We included 995 households in the final analysis.

### Magnitude of OOP expenditures

The mean total OOP medical expenditures (in 2021 USD) were $5·6 (95% CI: $4·3, 6.8), $7·8 ($5·3, 10·3), $9·0 ($6·4, 11·6), and $7·4 ($3·0, 11·9) for outpatient diarrhea, pneumonia, pertussis, and measles care, respectively. DMEs constituted 55% to 64% of total OOP expenditures, while DNMEs, mainly transportation expenditures, contributed 36% to 45% of total OOP expenditures. Drug and supply expenditures were major drivers amounting to 43% (36% to 47% by condition) of outpatient expenditures. OOP expenditures were higher for inpatient care at $65·3 (95% CI: $24·3, 106·2) for severe diarrhea, $51·6 ($32·0, 71·1) for severe pneumonia, $47·8 ($28·6, 67·0) for severe pertussis, $40·6 ($12·9, 68·3) for severe measles, and $101·7 ($88·5, 114·8) for meningitis. DME comprised 50% to 73% of total OOP expenditures, while drugs and supplies represented about 35%, followed by laboratory expenditures (around 17%), and expenditures for hospital stay (around 10%). Furthermore, DNME constituted 27% to

**Table 1. Summary characteristics of the study sample, by type of disease episode.**

| | Pneumonia | Diarrhea | Meningitis | Pertussis | Measles |
|---|---|---|---|---|---|
| **Total number of cases**[a] (inpatient cases in parentheses) | 327 (42) | 318 (31) | (252) | 60 (12) | 38 (8) |
| Oromia | 148 (15) | 150 (14) | 95 (95) | 29 (3) | 10 (4) |
| SNNP | 74 (8) | 69 (6) | 53 (53) | 7 (2) | 8 (3) |
| Amhara | 62 (12) | 61 (4) | 33 (33) | 15 (5) | 16 (1) |
| Somali | 18 (2) | 18 (2) | 23 (23) | -- | -- |
| Afar | 15 (3) | 14 (2) | 24 (24) | 4 (1) | 1 (0) |
| Addis Ababa | 10 (2) | 6 (3) | 24 (24) | 5 (1) | 3 (0) |
| Mean age, in years (95% CI) | 1·7 (1·5, 1·8) | 1·7 (1·6, 1·8) | 2·3 (1·9, 2·7) | 1·7 (1·4, 2·1) | 2·4 (1·7, 3·0) |
| Percentage of female children | 45% (146) | 48% (152) | 40% (101) | 43% (26) | 42% (16) |
| Percentage of rural residents | 46% (152) | 40% (128) | 38% (97) | 45% (27) | 18% (7) |
| Mean duration of hospitalization (in days)[b] (95% CI) | 4·5 (3·9, 5·1) | 4·0 (3·4, 4·5) | 10·6 (10·1, 11·2) | 4·8 (4·0, 5·7) | 4·4 (2·6, 6·2) |
| Mean number of people per household (95% CI) | 4·9 (4·7, 5·1) | 4·7 (4·5, 4·9) | 5·1 (4·8, 5·3) | 5·2 (4·7, 5·7) | 4·4 (4·0, 4·9) |
| Percentage (number) of respondents who were mother | 72% (234) | 74% (234) | 58% (145) | 68% (41) | 79% (30) |
| Percentage (number) of respondents who were father | 26% (85) | 25% (79) | 40% (100) | 30% (18) | 21% (8) |
| Mean respondent age, in years (95% CI) | 30 (29, 31) | 29 (28, 29) | 32 (31, 33) | 31 (29, 33) | 30 (28, 33) |
| Percentage (number) of respondents with secondary education or more | 31% (103) | 35% (112) | 35% (87) | 23% (14) | 29% (11) |
| Percentage (number) of respondents with paid employment | 28% (91) | 26% (81) | 47% (119) | 28% (17) | 32% (12) |

CI, confidence interval; SNNP, Southern Nations, Nationalities, and Peoples.

[a]This refers to the total number of cases in all the regions.

[b]For severe cases requiring inpatient care.

50% of total OOP expenditures; food and lodging (about 25%) were major drivers, followed by transportation expenditures (12%) (Fig 1).

The outpatient expenditures (per episode) were 3 and 13 times higher in hospitals compared with health centers and posts, respectively (Table 2). The type of health facility visited was statistically associated with a difference in OOP expenditures. Despite the effort by the government to provide care at health posts free of charge [36], households incurred important expenditures and drug expenditures contributed about 52% (42% to 60% depending on condition) of total expenditures. Other predictors were wealth and region (Table 2).

The mean (SD) wage losses for caregivers related to outpatient and inpatient care for the 5 VPDs combined were $2·3 ($11.5) and $23·3 ($52.1) (per illness episode), respectively; those ranged from $1·3 ($7.7) for diarrhea to $2·9 ($15.4) for pneumonia for outpatient care, and from $3·9 ($5.8) for measles to $26·7 ($55.7) for meningitis inpatient care (Table 3).

### Cases of CHE and IHE

The mean household annual consumption expenditures and nonfood expenditures were $1,944 and $819, respectively. The consumption expenditures varied substantially by wealth

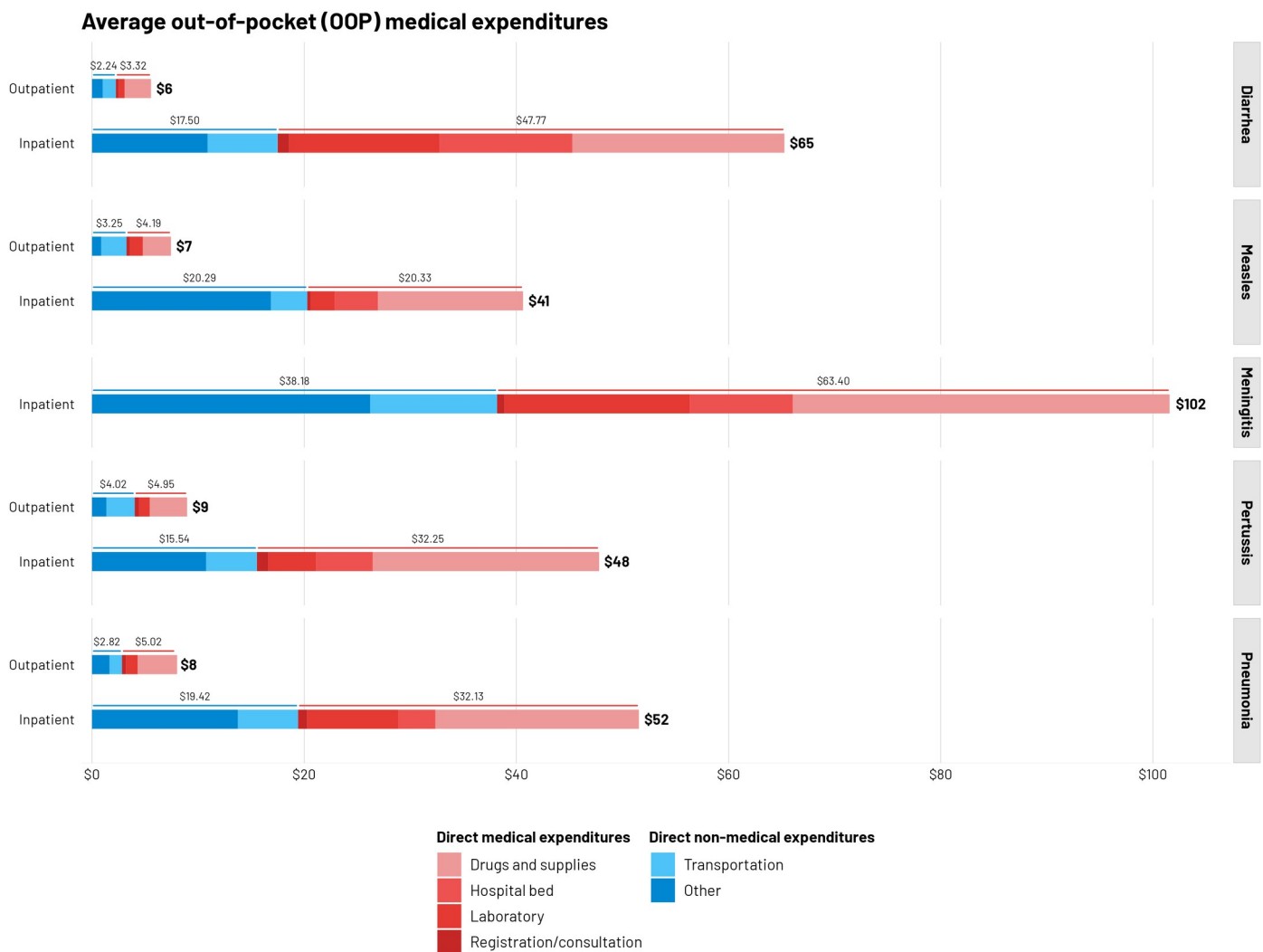

**Fig 1. Mean out-of-pocket medical expenditures (USD), per disease episode, by expenditure type and diagnosis.**

**Table 2. Association between OOP medical expenditures and wealth quintile, area of residence, facility type, and region, for either inpatient or outpatient cases (for all conditions combined).**

|  |  | Outpatient cases | | Inpatient cases | |
|---|---|---|---|---|---|
|  |  | Mean (95% CI) OOP expenditures | P value[a] | Mean (95% CI) OOP expenditures | P value[a] |
| Wealth quintile | I (poorest) | 4·6 (3·1, 6·2) |  | 63·2 (40·4, 85·9) |  |
|  | II | 5·7 (4·2, 7·1) | 0·60 | 80·6 (61·8, 99·5) | 0·66 |
|  | III | 6·6 (4·4, 8·7) | 0·49 | 81·7 (52·4, 111·1) | 0·98 |
|  | IV | 6·4 (4·0, 8·8) | 0·29 | 90·9 (69·0, 112·8) | 0·42 |
|  | V (Richest) | 12·9 (6·7, 19·1) | <0·001[b] | 111·5 (88·3, 134·7) | 0·03[b] |
| Area of residence | Urban | 7·7 (5·9, 9·4) | 0·86 | 98·5 (83·8, 113·3) | 0·20 |
|  | Rural | 5·9 (4·2, 7·7) |  | 72·8 (58·6, 87·1) |  |
| Facility type | Hospital | 13·2 (10·1, 16·3) |  | 89·0 (78·3, 99·7) | NA |
|  | HC | 4·1 (3·0, 5·2) | <0·001[c] | --- |  |
|  | HP | 1·0 (0·3, 1·6) | <0·001[c] | --- |  |
| Region | Addis Ababa | 18·6 (13·4, 23·8) |  | 166·5 (110·7, 222·3) |  |
|  | Afar | 25·3 (9·8, 40·8) | 0·10 | 81·9 (65·5, 98·3) | <0·001[e] |
|  | Amhara | 4·3 (2·9, 5·6) | 0·011[d] | 62·8 (42·1, 83·4) | <0·001[e] |
|  | Oromia | 5·5 (3·9, 7·0) | 0·024[d] | 78·2 (58·6, 97·8) | <0·001[e] |
|  | SNNP | 4·4 (3·2, 5·6) | 0·027[d] | 75·6 (64·5, 86·8) | <0·001[e] |
|  | Somali | 19·4 (6·8, 32·0) | 0·054 | 152·3 (107·1, 197·5) | 0·515 |

[a]P values are based on linear regression analysis using the outcome variable "OOP expenditures" and the independent variables "wealth quintile," "area of residence," "facility type," and "region." The poorest quintile, urban residence, hospital visits, Addis Ababa were reference values against which comparisons were made while adjusting for all the independent variables.

[b]The difference in OOP medical expenditures was only significant between the poorest (I) and the richest (V) quintiles.

[c]The expenditures in hospitals were significantly higher than in HCs and HPs.

[d]The expenditures in Addis Ababa were significantly different from those in Amhara, Oromia, and SNNP.

[e]The expenditures in Addis Ababa were significantly different from those in Afar, Amhara, Oromia, and SNNP.

HC, health center; HP, health post; OOP, out-of-pocket; SNNP, Southern Nations, Nationalities, and Peoples.

quintile: The richest spent 4 times more compared with the poorest ($3,639 versus $907). Food expenditures accounted for 68% of total expenditures for the poorest (48% for the richest). For outpatient care, 0·2% and 0·3% incurred CHE with a 10% threshold of annual expenditures

**Table 3. Estimated mean income or wage losses (in 2021 USD) per illness episode for combined inpatient and outpatient care by disease type and region.**

|  | Mean (SD) wage loss ($): Diarrhea | Mean (SD) wage loss ($): Measles | Mean (SD) wage loss ($): Meningitis | Mean (SD) wage loss ($): Pertussis | Mean (SD) wage loss ($): Pneumonia |
|---|---|---|---|---|---|
| Average loss[a] | 3·61 (21·24) | 1·86 (4·80) | 26·66 (55·74) | 2·74 (7·59) | 4·31 (17·01) |
| Addis Ababa | 69·51 (136·55) | ---- | 24·96 (35·65) | 10·60 (20·06) | 5·30 (9·90) |
| Afar | 16·52 (25·72) | 23·04[b] | 28·14 (32·25) | 14·67 (9·86) | 34·87 (62·80) |
| Somali | 5·59 (14·24) | --- | 71·63 (127·37) | --- | 8·38 (17·93) |
| Amhara | 3·13 (7·87) | 0·72 (2·01) | 21·62 (42·92) | 2·57 (4·36) | 3·00 (6·99) |
| Oromia | 0·64 (4·50) | 1·15 (3·64) | 18·66 (43·99) | 0·06 (0·34) | 1·27 (6·89) |
| SNNP | 1·62 (3·65) | 3·07 (5·00) | 24·72 (37·99) | 1·81 (4·79) | 4·17 (9·83) |

[a]Average loss refers to the average values for the 5 regions and Addis Ababa city administration combined.

[b]No SD is reported as only one observation was estimated.

—No data were collected.

SD, standard deviation; SNNP, Southern Nations, Nationalities, and Peoples.

versus a 40% threshold of annual capacity to pay. Expectedly, the CHE estimates would become much higher with monthly consumption expenditures or capacity to pay used as denominators and when a lower CHE threshold (for example, 5% annual consumption expenditure threshold) are used. Similarly, higher CHE incidence was observed for inpatient care (Table 4).

Only 2% of households interviewed were below the poverty line ($1·90 per day, 2020 PPP), compared with the national average of 31% in 2015 (2011 PPP) [3]. The Pen's Parade graph (Fig 2) shows households below the poverty line before incurred OOP expenditures and households pushed below it with incurred OOP expenditures. For outpatient care, 1.5% (from 1.1% for diarrhea to 3.3% for measles) of households were impoverished; for inpatient care, it was much higher with 32% on average, from 14% for pneumonia to 36% for meningitis (Table 4).

The findings from the logistic regressions show that the odds of CHE were 13 times higher among those patients who received inpatient care and 10 times higher among those patients who sought care at hospitals (versus health centers). Other factors significantly associated with CHE were meningitis and lower wealth (Table 5).

Lastly, households used various mechanisms to cope with high OOP payments. Current income in 423 households (about 45% of the time) and savings in 202 households (22%) were most common. The other commonly used coping strategies included asset sales in 125 households (13%), borrowing in 69 households (7%) and support from family and/or friends in 36 households (4%). Reimbursement by insurance occurred only 8% (76 households) of the time.

## Discussion

We quantified OOP expenditures and wage losses associated with 5 common VPDs in Ethiopia. The mean total outpatient OOP medical expenditures (in 2021 USD) ranged from $5·6

**Table 4. Estimated incidence of CHEs and IHEs by disease category.**

CHE incidence, at 10% of monthly consumption expenditures and 40% of monthly capacity to pay

|  | Diarrhea | | Pneumonia | | Pertussis | | Measles | | Meningitis |
|---|---|---|---|---|---|---|---|---|---|
|  | OP | IP | OP | IP | OP | IP | OP | IP | IP |
| 10% threshold | 12·9% (37/287) | 90·3% (28/31) | 14·7% (42/285) | 81·2% (34/42) | 33·3% (16/48) | 91·7% (11/12) | 23·3% (7/30) | 100% (8/8) | 94·1% (237/252) |
| 40% threshold | 9·8% (28/287) | 77·4% (24/31) | 12·6% (36/285) | 78·6% (33/42) | 22·9% (11/48) | 83·3% (10/12) | 13·3% (4/30) | 87·5% (7/8) | 88·9% (224/252) |

CHE incidence, at 5% and 10% of annual consumption expenditures and 40% of annual capacity to pay

|  | Diarrhea | | Pneumonia | | Pertussis | | Measles | | Meningitis |
|---|---|---|---|---|---|---|---|---|---|
|  | OP | IP | OP | IP | OP | IP | OP | IP | IP |
| 5% threshold | 0.4% (1/287) | 22.6% (7/31) | 0.4% (1/285) | 14.3% (6/42) | 0.0% (0/48) | 25% (3/12) | 3.3% (1/30) | 12.5% (1/8) | 39.7% (100/252) |
| 10% threshold | 0·0% (0/287) | 12·9% (4/31) | 0·0% (0/285) | 7·1% (3/42) | 0·0% (0/48) | 0·0% (0/12) | 3·3% (1/30) | 0·0% (0/8) | 15·5% (39/252) |
| 40% threshold | 0·4% (1/287) | 9·7% (3/31) | 0·0% (0/285) | 11·9% (5/42) | 0·0% (0/48) | 8·3% (1/12) | 3·3% (1/30) | 0·0% (0/8) | 13·5% (34/252) |

IHE incidence

|  | Diarrhea | | Pneumonia | | Pertussis | | Measles | | Meningitis |
|---|---|---|---|---|---|---|---|---|---|
|  | OP | IP | OP | IP | OP | IP | OP | IP | IP |
| % pushed into poverty | 1·1% (3/287) | 29·0% (9/31) | 1·8% (5/285) | 14·3% (6/42) | 2·1% (1/48) | 25% (3/12) | 3·3% (1/30) | 25% (2/8) | 36·1% (91/252) |

CHE, catastrophic health expenditure; IHE, impoverishing health expenditure; IP, inpatient; OP, outpatient.

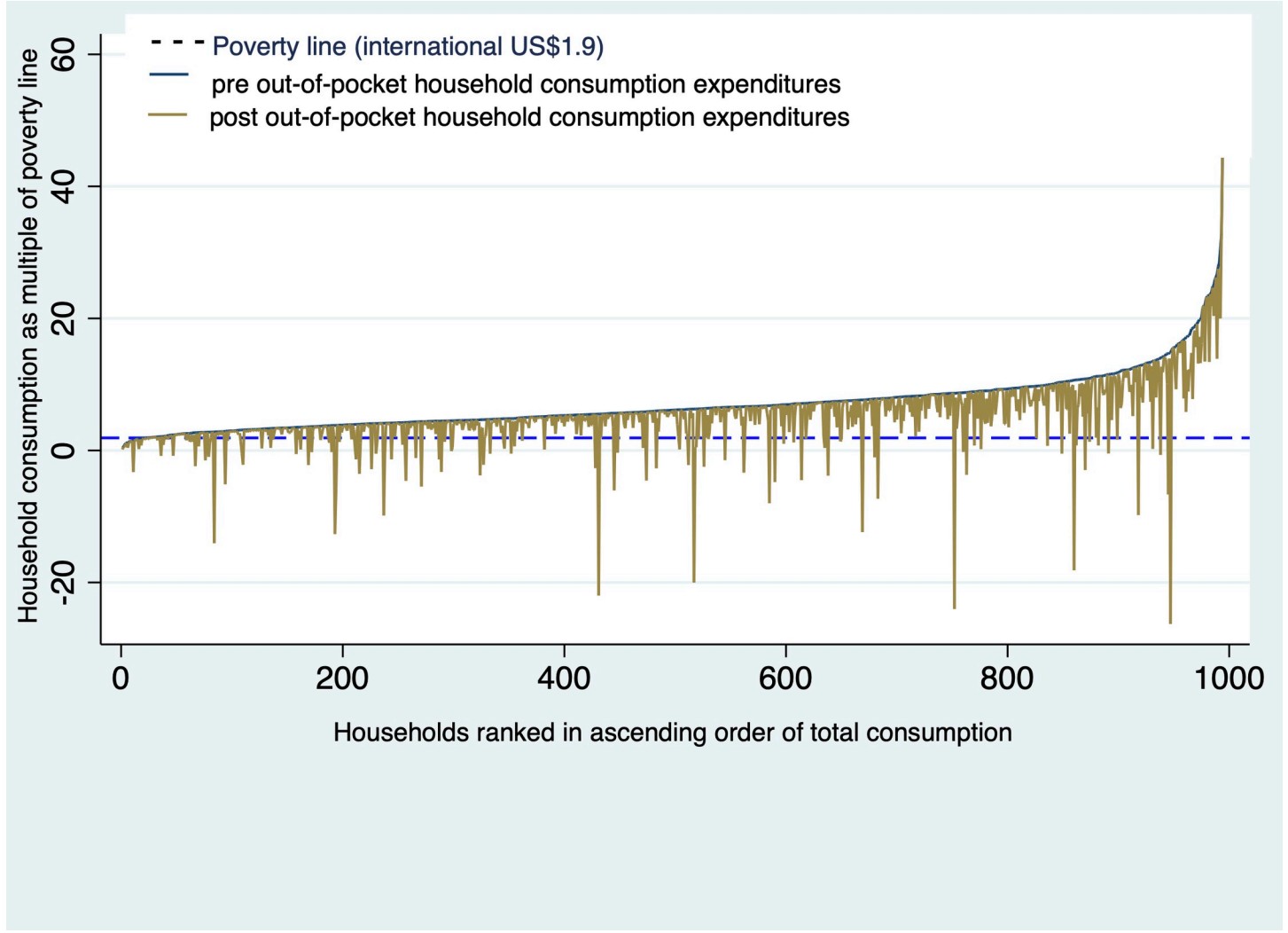

**Fig 2. Impact of out-of-pocket expenditures associated with treatment of vaccine-preventable diseases on monthly household consumption expenditures.**

(95% CI: $4·3, 6.8) for diarrhea to $9·0 ($6·4, 11·6) for pertussis; the expenditures for inpatient care were larger, ranging from $40·6 ($12·9, 68·3) for severe measles to $101·7 ($88·5, 114·8) for meningitis. Our findings demonstrate that care-seeking for VPDs can lead to a substantial financial burden for Ethiopian families. Most households (about 92%) did not have insurance coverage and incurred OOP expenditures at the point of care. DMEs were a major contributor to these OOP expenditures (ranging from 50% to 73% by condition). Similar to previous reports, for both inpatient and outpatient care, medications accounted for the largest share of the DMEs followed by (for inpatient care) laboratory costs and costs of hospital stay [13,14,37–40]. As for the DNMEs, transportation, food, and lodging costs were major drivers, particularly for inpatient care as families had to travel long distances to access hospitals. Additionally, important wage losses would also be incurred while taking care of sick children.

Type of health facility visited, economic status, and region were factors significantly associated with the level of OOP expenditures incurred. Higher expenditures would arise with hospitals, care-seeking in large cities, or for wealthier households; smaller expenditures would arise at health centers and health posts; and larger expenditures would be attributable to drug

**Table 5. Factors associated with odds of CHEs for VPDs.**

| Covariate | | Unadjusted | | Adjusted | | |
|---|---|---|---|---|---|---|
| | | Odds ratio | 95% CI | Odds ratio[a] | 95% CI | P value |
| Wealth quintile | I (poorest) | 2·21 | 1·22, 3·99 | 2·18 | 1·13, 4·22 | 0·02 |
| | II | 2·88 | 1·53, 5·42 | 3·23 | 1·67, 6·26 | <0·001 |
| | III | 1·43 | 0·88, 2·85 | 1·27 | 0·66, 2·45 | 0·47 |
| | IV | 1·63 | 0·93, 2·82 | 1·48 | 0·76, 2·89 | 0·25 |
| | V (richest) | Reference | | Reference | | |
| Area of residence | Rural | 1·25 | 0·97, 1·63 | 1·26 | 0·81, 1·97 | 0·31 |
| | Urban | Reference | | Reference | | |
| Facility type | Hospital | 34·56 | 22·32, 53·51 | 10·33 | 6·09, 17·51 | <0·001 |
| | Health center | Reference | | Reference | | |
| Type of care | Inpatient | 63·27 | 40·50, 98·85 | 13·97 | 7·00, 27·89 | <0·001 |
| | Outpatient | Reference | | Reference | | |
| VPD type | Meningitis | 61·50 | 34·13, 110·80 | 2·65 | 1·12, 6·27 | 0·03 |
| | Measles | 2·54 | 1·25, 5·14 | 2·02 | 0·78, 5·22 | 0·15 |
| | Pertussis | 3·18 | 1·79, 5·67 | 1·28 | 0·62, 2·65 | 0·50 |
| | Pneumonia | 1·18 | 0·81, 1·71 | 1·02 | 0·62, 1·66 | 0·95 |
| | Diarrhea | Reference | | Reference | | |
| Age of respondent | | 1·05 | 1·03, 1·07 | 0·99 | 0·96, 1·02 | 0·53 |
| Household family size | | 1·06 | 0·99, 1·13 | 1·05 | 0·93, 1·19 | 0·82 |

CHE, catastrophic health expenditure; CI, confidence interval; VPD, vaccine-preventable disease.

[a]The richest quintile, urban residence, health center visits, outpatient visits, diarrheal illness were reference values against which comparisons were made while adjusting for all the independent variables.

expenses. Clinical care activities for the management of childhood illnesses such as diarrhea, pneumonia, or measles are among the fee-exempted services as proposed in the 2019 Ethiopian essential health services package [41]. This points to the lack of an accompanying healthcare financing reform in Ethiopia.

The risk of CHE for outpatient care for VPDs was around 0·2% (at a 10% threshold of annual expenditures) and 0·3% (at a 40% threshold of capacity to pay), while much higher rates (from 3·3% to 15·5% by condition) were observed for inpatient care. Similarly, 14% to 36% of households seeking inpatient care for a sick child would be pushed below the poverty line. A previously published Ethiopian study reported an 11% CHE incidence and a 6% to 7% IHE incidence for seeking inpatient care for severe pneumonia and diarrhea among children aged under 5 years [13]. The current analysis is consistent while reporting slightly higher levels of IHE for inpatient care (possibly related to the lower poverty line of US$1.25 used in [13]). Expanding access to vaccination has the potential to protect families from OOP expenditures and substantially reduce CHE and IHE associated with VPDs where such benefits would primarily accrue among the poorest [20].

The likelihood of CHE varied by type of care, household economic status, type of facility visited, and disease; receiving inpatient care and hospital visits were the most important predictors of CHE. Poorer households had lower OOP expenses but were more likely to suffer from CHE as compared to their richer counterparts. As expected, inpatient care and hospital visits resulted in substantially higher financial catastrophes, which might be related to longer travel times to and hospital stays highlighting the importance of DNMCs to CHE. Tolla and colleagues had also reported similar observations in their study on OOP expenditures for cardiovascular diseases in Addis Ababa [42].

Households used various sources of financing besides current income and savings such as asset sales and borrowing, which might lead to detrimental long-term economic consequences. Poorer households were more likely to resort to such coping mechanisms, consistent with previous studies [42,43]. Our sample had only 8% of participants with insurance coverage despite a reported 37% community-based health insurance (CBHI) enrollment coverage for the years 2020/2021 among the population in the informal sector [44]. Local evidence, however, suggests a likely positive impact of CBHI on health services utilization and financial risk protection in Ethiopia [45].

Our findings are not without limitations. First, we could not collect the required sample size for measles and pertussis cases; therefore, estimates around inpatient care and disaggregated analysis for these 2 conditions should be interpreted with caution. In fact, data collection occurred a few months after measles supplemental immunization activities took place in the country, which might have led to fewer measles cases [46]. Second, our study did not account for those who did not seek care or used alternative sources of treatment, which might overestimate the incidence of CHE. Facility-based cross-sectional studies only capture families who use health services and thus could be biased towards urban and wealthier households. The fact that our sample had only 2% of the participants below the poverty line (far below the national average of around 31%) might illustrate such a bias. Most Ethiopian CBHI beneficiaries live within rural communities, yet our sample was biased towards urban residents, which might have resulted in underrepresentation of insurance beneficiaries. The data also indicate lower OOP expenses for rural residents for both outpatient and inpatient services as compared to their urban counterparts. Despite lower OOP expenses, rural households were more likely to incur CHE, possibly a function of rural households' inability to cope with medical payments. The underrepresentation of rural households in our study could underestimate the extent of CHE. Third, we prioritized facilities from likely high-burden areas to maximize recruitment of patients with VPDs. Even though we do not anticipate healthcare costs in public health facilities to differ based on the burden of VPDs, socioeconomic characteristics of the households may vary that could affect the study findings. Fourth, we included only public facilities, which may underestimate the extent of OOP expenditures due to the likely higher costs in private facilities. Lastly, data on OOP payments and household expenditures relied on self-reporting with a significant risk for reporting errors. However, collecting OOP data immediately upon occurrence might minimize recall bias.

The extent of CHE and IHE for VPDs, especially for those who require inpatient care, is very high in Ethiopia. We found the financial catastrophe would be more pronounced among the poorest populations. Ensuring equitable distribution of health services associated with high financial risk protection is a major objective of the Ethiopian health sector [26,47]. Despite remarkable progress in access to vaccines in Ethiopia, there are still large coverage gaps with substantially lower access for the poor who face higher risks of VPDs. For these households, VPDs increase not only morbidity and mortality but also the risk of financial hardship, which could worsen inequities and poverty in Ethiopia. Expanding equitable access to vaccines cannot be overemphasized from both health and economic standpoints. Such realization requires the government's commitment toward increasing and sustaining vaccine financing in Ethiopia.

## Supporting information

**S1 STROBE checklist. STROBE Statement—Checklist of items that should be included in reports of *cross-sectional studies*.**
(DOCX)

**S1 PROSPECTIVE protocol. Webappendix 2: Prospective protocol (abridged).**
(PDF)

**S1 Data Tool. Data collection tool.**
(PDF)

## Acknowledgments

We thank the healthcare providers who supported the data collection process. Our sincere appreciation also goes to parents of the children who volunteered to take part in the study. Finally, we are indebted to the Ethiopian Public Health Institute and the Maternal, Child Health, and Nutrition Directorate of the Ministry of Health, as well as to Abebe Bekele for their invaluable support. The views expressed are those of the authors and not necessarily those of the funder.

## Author Contributions

**Conceptualization:** Solomon Tessema Memirie, Mieraf Taddesse Tolla, Eva Rumpler, Stéphane Verguet.

**Data curation:** Solomon Tessema Memirie.

**Formal analysis:** Solomon Tessema Memirie.

**Funding acquisition:** Stéphane Verguet.

**Investigation:** Solomon Tessema Memirie, Stéphane Verguet.

**Methodology:** Solomon Tessema Memirie, Mieraf Taddesse Tolla, Eva Rumpler, Stéphane Verguet.

**Project administration:** Solomon Tessema Memirie.

**Supervision:** Solomon Tessema Memirie.

**Validation:** Solomon Tessema Memirie, Stéphane Verguet.

**Visualization:** Solomon Tessema Memirie, Sarah Bolongaita.

**Writing – original draft:** Solomon Tessema Memirie.

**Writing – review & editing:** Solomon Tessema Memirie, Mieraf Taddesse Tolla, Eva Rumpler, Ryoko Sato, Sarah Bolongaita, Yohannes Lakew Tefera, Latera Tesfaye, Meseret Zelalem Tadesse, Fentabil Getnet, Tewodaj Mengistu, Stéphane Verguet.

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
