## [Editor Report · Decision Letter 0]

1 Aug 2022

Dear Dr Memirie, 

Thank you for submitting your manuscript entitled "Out-of-pocket expenditures and financial risks associated with treatment of vaccine-preventable diseases in Ethiopia" for consideration by PLOS Medicine.

Your manuscript has now been evaluated by the PLOS Medicine editorial staff and I am writing to let you know that we would like to send your submission out for external peer review.

Please re-submit your manuscript within two working days, i.e. by Aug 03 2022 11:59PM.

Kind regards,

Beryne Odeny

PLOS Medicine

---

## [Decision Letter · Decision Letter 1]

18 Oct 2022

Dear Dr. Memirie,

Thank you very much for submitting your manuscript "Out-of-pocket expenditures and financial risks associated with treatment of vaccine-preventable diseases in Ethiopia" (PMEDICINE-D-22-02567R1) for consideration at PLOS Medicine. 

[LINK]

In light of these reviews, I am afraid that we will not be able to accept the manuscript for publication in the journal in its current form, but we would like to consider a revised version that addresses the reviewers' and editors' comments. Obviously we cannot make any decision about publication until we have seen the revised manuscript and your response, and we plan to seek re-review by one or more of the reviewers. 

We expect to receive your revised manuscript by Nov 08 2022 11:59PM. Please email us (plosmedicine@plos.org) if you have any questions or concerns.

We look forward to receiving your revised manuscript. 

Sincerely,

Philippa Dodd MBBS MRCP PhD

PLOS Medicine

plosmedicine.org

GENERAL

Please address all editor and reviewer comments detailed below, in full

Your study combines observational and economic/cost analyses. Please review the following relevant guidance and report your study according to relevant reporting guidelines. I have made some suggestions below.

Please ensure that the study is reported according to the STROBE guideline, and include the completed STROBE checklist as Supporting Information. Please add the following statement, or similar, to the Methods: "This study is reported as per the Strengthening the Reporting of Observational Studies in Epidemiology (STROBE) guideline (S1 Checklist)." The STROBE guideline can be found here: http://www.equator-network.org/reporting-guidelines/strobe/ When completing the checklist, please use section and paragraph numbers, rather than page numbers.

Please ensure that you consider the following guidance, relevant to your study design, when reporting your cost analysis, provided at:

http://www.equator-network.org/post_type=eq_guidelines&eq_guidelines_study_design=economic-evaluations&eq_guidelines_clinical_specialty=0&eq_guidelines_report_section=0&s=

Please also provide the relevant completed checklist. In the checklist please include sufficient text excerpted from the manuscript to explain how you accomplished all applicable items.

ABSTRACT

Please structure your abstract using the PLOS Medicine headings (Background, Methods and Findings, Conclusions). Please combine the Methods and Findings sections into one section, “Methods and findings”.

Abstract Methods and Findings:

Please ensure that all numbers presented in the abstract are present and identical to numbers presented in the main manuscript text.

Please explicitly state when data was collected from households and over what time frame if there was more than one data collection point (please include mean, SD and range)

Please explicitly state the main outcome measures 

Please quantify the main results with 95% CIs and p-values

Please include any important dependent variables that are adjusted for in the analyses

Would it be helpful to include the number of participants in the 995 households? I’m assuming 1 child per household (so 995 children) but these are contagious VPDs and households usually have more than one child so that is quite a big assumption to make (this is only explicitly clear by the time the reader reaches the results section of the main manuscript)

Line 46: “..OOP costs” perhaps replace with expenditure for consistency? Please check and amend throughout (including in the main manuscript)

Line 55: “…13·3% of households…” please include denominator, for example, “…13.3% of 995 households…” or something similar

In the last sentence of the Abstract Methods and Findings section, please describe the main limitation(s) of the study's methodology.

Abstract Conclusion

Line 58: “Conclusions: The OOP costs induced by VPDs are large in Ethiopia, especially for the poor and for patients requiring inpatient care.” Suggest “Conclusions: The OOP costs induced by VPDs are substantial in Ethiopia and disproportionately impact those with low-incomes and those requiring inpatient care.” Or something similar

AUTHOR SUMMARY

METHODS and RESULTS

Please report the number of patients, households, etc and dates of recruitment

Please account for all methods used in your study - please discuss how the data pertaining to wage-loss were collected/calculated in the methods section of the manuscript (see also reviewer comments below)

Please define the length of follow up (eg, in mean, SD, and range).

Please present numerators and denominators for percentages, at least in the Tables (see also comments below)

Please define "lost to follow-up" as used in this study. Other reasons for exclusion should be defined.

Please specify whether informed consent was written or oral.

Did your study have a prospective protocol or analysis plan? Please state this (either way) early in the Methods section. 

For all observational studies, in the manuscript text, please ensure that you have clearly indicated: 

(1) the specific hypotheses you intended to test, 

(2) the analytical methods by which you planned to test them,

(3) the analyses you actually performed, and 

(4) when reported analyses differ from those that were planned, transparent explanations for differences that affect the reliability of the study's results. If a reported analysis was performed based on an interesting but unanticipated pattern in the data, please be clear that the analysis was data-driven.

Line 235: “We employed face-to-face exit interviews…” this statement implies that there was an entry interview – my understanding is that this was the initial interview – for clarity perhaps instead “We employed face-to-face interviews

TABLES

Table 1: please present denominators for percentages reported here. For the row labelled: Mean household size – consider “Mean number of people per household”

Table 2: It’s not clear on only reviewing the figure what the “mean” numbers here refer to, please amend the table for improved clarity and accessibility to the reader. Similarly which piece of data presented do these p-values refer to? Please also include the statistical tests used to determine these. were any factors adjusted for in these analyses? If so please specify which in the figure caption and include the unadjusted analyses. Where p-values are reported please also report 95% CIs. In the figure caption I would suggest replacing these symbols with superscript letters a, b, c and so on

Table 3; “Average Loss” please add USD symbol in parentheses to make it clear what these numbers refer to. Please remove “(standard deviation in parentheses)” from the table title and place in the caption/legend below

Table 4: Please present numerators and denominators for percentages, at least in the Tables [not necessarily each time they're mentioned].

Table 5: were any factors adjusted for in these analyses? If so please specify which in the figure caption and include the unadjusted analyses

FIGURES

Throughout, to make the figures more accessible to those with colour blindness please avoid the use of red and/or green

Figure 1 title: “…out-of-pocket medical expenditures” see previous comments re: Costs Vs Expenditure – please amend accordingly

Figure 2: please define - HH, OOP and (Int. US$)

*** Throughout the manuscript there is inconsistent use of the terms “Expenditure” and “cost(s)” in figure and table titles and legends/captions please check and revise using one of these terms only ***

DISCUSSION

Please ensure that your discussion is structured as follows: a short, clear summary of the article's findings; what the study adds to existing research and where and why the results may differ from previous research; strengths and limitations of the study; implications and next steps for research, clinical practice, and/or public policy; one-paragraph conclusion.

Please remove the funding statement and declaration of interests statement from the end of the manuscript and include only in the manuscript submission form 

REFERENCES

For in text reference call outs, citations should be in square brackets, and preceding punctuation. Note the absence of spaces within square bracket, for example: – “asymptomatically [2,4].”

In the reference list, please ensure that no more than the first 6 authors are listed before et al where more than 6 authors have contributed to a study including in the supplementary references

Journal name abbreviations should be those found in the National Center for Biotechnology Information (NCBI) databases.

Please see our website for other reference guidelines https://journals.plos.org/plosmedicine/s/submission-guidelines#loc-references

SUPPORTING INFORMATION

Please include the S1 checklist in-line with STROBE reporting and any other checklists from reporting guidance relevant to your cost analyses

Comments from the reviewers:

Reviewer #1: The manuscript that aims to estimate the extend of out-of-pocket payment and the associated catastrophic health expenditures due to selected VPDs in Ethiopia is well written and of potential interested to the readers of the journal. I recommend that paper with proposed revisions should be considered for publication.

Major comments 

- In the introduction or discussion section the authors should elaborate on the policy relevance of these types of studies in general and who the target audience is e.g. the cost could be part of an input for return of investment analyses of vaccines and immunization programmes targeting Ministries of Finance.

- In the discussion section the authors rightly highlight the study is biased towards urban well off study population. Would it be feasible to extrapolate the study results if more marginalized populations from rural area would have included in the study? If a quantification is not feasible, an analysis describing the direction of main parameters influencing the OOP and CHE results would be useful for decision makers and researchers in the field.

Minor comments

- p 3. I suggest add the following key-words: costs (or costing)

- p10, line 171. I suggest to also add average life expectancy for male and female in Ethiopia

- p15 table 1. I guess the title of first column should be: Total number of cases by "city administration"

- p19 line 375-376: the 10% vs 40% CHE threshold of OOP should be introduced in the methods section including the rationale behind the "usually used" 10% and why 40%? Is there any rationale? Why not adding a lower bound of 5% to give the full range of the sensitivity of CHE depending on the threshold?

Reviewer #2: This study conducted a costing analysis trying to estimate the out-of-pocket (OOP) expenditure and productivity losses due to five vaccine-preventable diseases (VPDs) and assess the catastrophic health expenditures (CHE) caused by these VPDs. Overall, I think this study is well designed and conducted. The topic of the study is of interest and the results are very helpful to promote vaccination coverage in the low-and-middle income countries. Below are my specific comments. 

1. "The link to the public data repository will be available after acceptance of the manuscript." I think the data repository needs to be made available before acceptance of the paper. However, I am not sure whether that would breach the confidentiality of the personal data. 

2. Line 143-144: would you specify the SDGs that vaccine can help realize? 

3. Line 188: how the two hospitals were selected? Were they selected randomly? 

4. Line 181-209: Is there a valid reason why you purposefully selected areas with high burden of the VPDs in the country? Will that selection bias your results?

5. Line 208-209: I understand public facilities used by most people. However, the 25% care seeking that takes place in private section may cost much more. Therefore, excluding those may severely underestimate the cost of seeking care for the VPDs on the national level. 

6. Line 217: How were the mean 15 USD and SD of 32 USD were estimated or chosen?

7. Line 296: ETB 8401 per year? 

8. Line 301-302: Is that linear or logistic regression that you used? It's a bit confusing to me.

Reviewer #3: Dear Authors:

Thank you very much for the opportunity to review your manuscript entitled Out-of-pocket expenditures and financial risks associated with treatment of vaccine preventable diseases in Ethiopia. Your research paper explored out-of-pocket expenditures and financial risks of vaccine-preventable diseases from house-hold perspectives.

The paper is well-written with some room for improvement. Please see below for my comments.

Line 194: The sample was taken from the high burden areas. Does this selection method bias the findings? Line 468 discusses some potential biases but the discussion is very limited to understand how the sampling technics might have contributed to biased results (or not). I recommend additional discussion on this.

Line 411: The discussion section does not include your assessment of vaccine's potential contributions to alleviate financial burdens. How would vaccination programs improve CHEs in percentages? (i.e. how would Table 4 look like after an implementation?) Are vaccination programs improve equity to health? If so, how would Table 2 look like once a program is implemented? While concrete numbers might not be found and out of scope for this study, I recommend some discussion regarding to return on investment for public policy stakeholders. 

Line 272: Wage-loss is first discussed in data analytics section. Please discuss how the data pertaining to this analysis are collected in the data collection section earlier in the manuscript. I suggest updating objectives to include measures used to identify productivity losses.

[LINK]

---

## [Decision Letter · Decision Letter 2]

3 Feb 2023

Dear Dr. Memirie,

Thank you very much for re-submitting your manuscript "Out-of-pocket expenditures and financial risks associated with treatment of vaccine-preventable diseases in Ethiopia" (PMEDICINE-D-22-02567R2) for review by PLOS Medicine.

I have discussed the paper with my colleagues and the academic editor and it was also seen again by 2 reviewers. I am pleased to say that provided the remaining editorial and production issues are dealt with we are planning to accept the paper for publication in the journal.

[LINK]

We look forward to receiving the revised manuscript by Feb 10 2023 11:59PM.   

Sincerely,

Philippa Dodd, MBBS MRCP PhD

PLOS Medicine

plosmedicine.org

Requests from Editors:

GENERAL

Thank you for your detailed and considerate responses to previous editor and reviewer comments. Please see below for further minor revisions which we require that you address, in full.

TITLE

Please revise your title according to PLOS medicine’s style. Your title must be nondeclarative and not a question. It should begin with main concept if possible. The study design ("A randomized controlled trial," "A retrospective study," "A modelling study," etc.) should be placed in the subtitle (ie, after a colon). Suggest – “Out-of-pocket expenditures and financial risks associated with treatment of vaccine preventable diseases in Ethiopia: a cross-sectional costing analysis.” Or something similar.

STATISTICAL REPORTING

Throughout, consider the use of commas to separate upper and lower confidence limits instead of hyphens (as these can be confused with the presentation of negative values). 

METHODS and RESULTS

Lines 285 onwards – we agree with the reviewer (see below) that this aspect of your methodology requires additional detail, please revise accordingly.

TABLES

Thank you for revising your tables which are now much clearer. 

To help facilitate transparent data reporting, PLOS Medicine requests that where adjusted analyses are reported, the unadjusted analyses are reported for comparison. Please include and if not please clearly state why not.

Table 1 – to improve accessibility to the reader please adjust the table so that confidence limits are reported on a single line

Table 2 – your footnote key begins with the letter “c” but logically should begin with the letter “a” please revise 

Table 3 – as above, your footnote key begins with the letter “h” and should begin with the letter “a”. Please revise throughout all tables 

Table 5 – as above, please replace “j” with “a”

SOCIAL MEDIA

To help us extend the reach of your research, please provide any Twitter handle(s) that would be appropriate to tag, including your own, your coauthors’, your institution, funder, or lab. Please detail any handles you wish to be included when we tweet this paper, in the manuscript submission form when you re-submit the manuscript.

Comments from Reviewers:

Reviewer #2: I think the authors have answered and addressed my and other reviewers' questions adequately. I believe the paper is acceptable for publication. Great work and congratulations to the authors! 

Reviewer #3: Dear Authors:

Thank you very much for the opportunity to review your manuscript entitled Out-of-pocket expenditures 

and financial risks associated with treatment of vaccine preventable diseases in Ethiopia once again. Your research 

paper explored out-of-pocket expenditures and financial risks of vaccine-preventable diseases from household perspectives.

The paper is well-written, and the authors provided detailed responses to the previous comments from reviewers and the editor.

I found one incomplete revision in response to my previous comment. Please address my additional request below:

Lines 285- 287, Thank you very much for adding the sentence to explain that data on productivity losses were collected. However, the added sentence does not explain how the data were collected. Please include the method. Did interviewers directly ask this question? Please explain the data collection method. This applies to comments from the editor on "methods and results". Just stating "data were collected" do not sufficiently explain how exactly data collection was conducted. While the study protocol can be referred to, a concise explanation in the manuscript itself would be a great improvement in transparency. 

Best regards,

Your colleague

[LINK]

---

## [Editor Report · Decision Letter 3]

10 Feb 2023

Dear Dr Memirie, 

On behalf of my colleagues and the Academic Editor, Dr. Rebecca Freeman-Grais, I am pleased to inform you that we have agreed to publish your manuscript "Out-of-pocket expenditures and financial risks associated with treatment of vaccine-preventable diseases in Ethiopia: a cross-sectional costing analysis " (PMEDICINE-D-22-02567R3) in PLOS Medicine.

Prior to publication we require that you make the following amendments to your manuscript:

1) Please amend the STROBE checklist and refer to section and paragraph numbers, not page (or line) numbers as these often change at the time of publication

2) Abstract line 61 - please report p values as p<0.001 or where higher the exact p value as p=0.02, for example. Please check and amend throughout the manuscript where relevant.

PRESS

Best wishes, 

Philippa Dodd, MBBS MRCP PhD 

PLOS Medicine